# RETHINKING MODALITY ALIGNMENT IN MULTI-MODAL LARGE LANGUAGE MODELS

## ABSTRACT

Multi-modal Large Language Models (MLLMs) demonstrate remarkable proficiency in addressing a wide range of Vision-Language (VL) tasks. However, most advancements have been focused on adapting to longer sequences containing detailed visual information and scaling up high-quality VL corpus. Prevalent VL alignment modules (e.g., the adapter layer in LLaVA and the Q-former in BLIP-2) struggle to align the LLM and visual inputs adequately. They rely on the powerful LLM to decode sub-optimally aligned visual features into the desired formatted word sequences, which can result in hallucinations and reduce the reliability of visual reasoning. Additionally, the LLM's causal attention does not effectively capture the relationship between visual embeddings. To tackle these issues, we rethink the modality alignment in MLLMs and present VL Superior Alignment (VLSA), a framework designed to decouple the alignment of the LLM with visual inputs. VLSA has two main stages: The **perception alignment** stage, which consists of innovative compressive high-resolution image encoding and reconstructive training based on Latent Diffusion Models (LDM), reduces the information loss in visual encoding and better models the spatial connection between images' subgraphs. The **cognition alignment** stage strengthens the LLM in understanding high-level visual semantics and low-level image appearances simultaneously. This advancement is actualized by following the instructions of predicting the codebook indices generated from a Vector Quantized (VQ) encoder and the pixel values within designated areas. Extensive experiments across 20 MLLM benchmarks underscore the consistent improvements brought by VLSA, demonstrating the effectiveness of our methods. In service to the MLLM research community, our code and model checkpoints will be publicly available.

## 1 INTRODUCTION

Large language models (LLMs) are advanced tools for processing, understanding, and generating contextual information. They act as powerful knowledge bases, providing valuable insights and enabling the creation of new content Dai et al. (2019); Devlin et al. (2018); Radford et al. (2019); Raffel et al. (2020); Zhang et al. (2022). As they continue to rapidly advance cha (2023); Zhang et al. (2022); Touvron et al. (2023), LLMs are increasingly integrating visual information to tackle Vision-Language (VL) tasks, including visual question answering (VQA) Antol et al. (2015); Singh et al. (2019) and image captioning (CAP) Chen et al. (2015); Sidorov et al. (2020). They have demonstrated notable progress Li et al. (2022a; 2021; 2022b); Wang et al. (2022a;b); Yang et al. (2021), surpassing traditional VL models.

In order to transform language-only LLMs into powerful multi-modal large language models (MLLMs), current techniques Bai et al. (2023); Alayrac et al. (2022); Li et al. (2023d); Gao et al. (2023); Liu et al. (2023b) generally adhere to a standard process involving using a pre-trained image encoder (such as CLIP Radford et al. (2021), or SigLIP Zhai et al. (2023)) to embed visual context, then integrating the visual and textual embeddings into LLMs for a range of tasks. These techniques can be categorized into two groups based on their VL integration methods: cross-attention-based integration (e.g., Flamingo Alayrac et al. (2022)) and concatenation-based integration (e.g., LLaVA Liu et al. (2023b)). Most prevalent MLLMs tend to favor the latter method under the paradigm of fine-tuning LLMs as it fully leverages the advanced capabilities of LLMs by consolidating multi-modal inputs into a comprehensive sequence and facilitating an equal treatment of both visual and

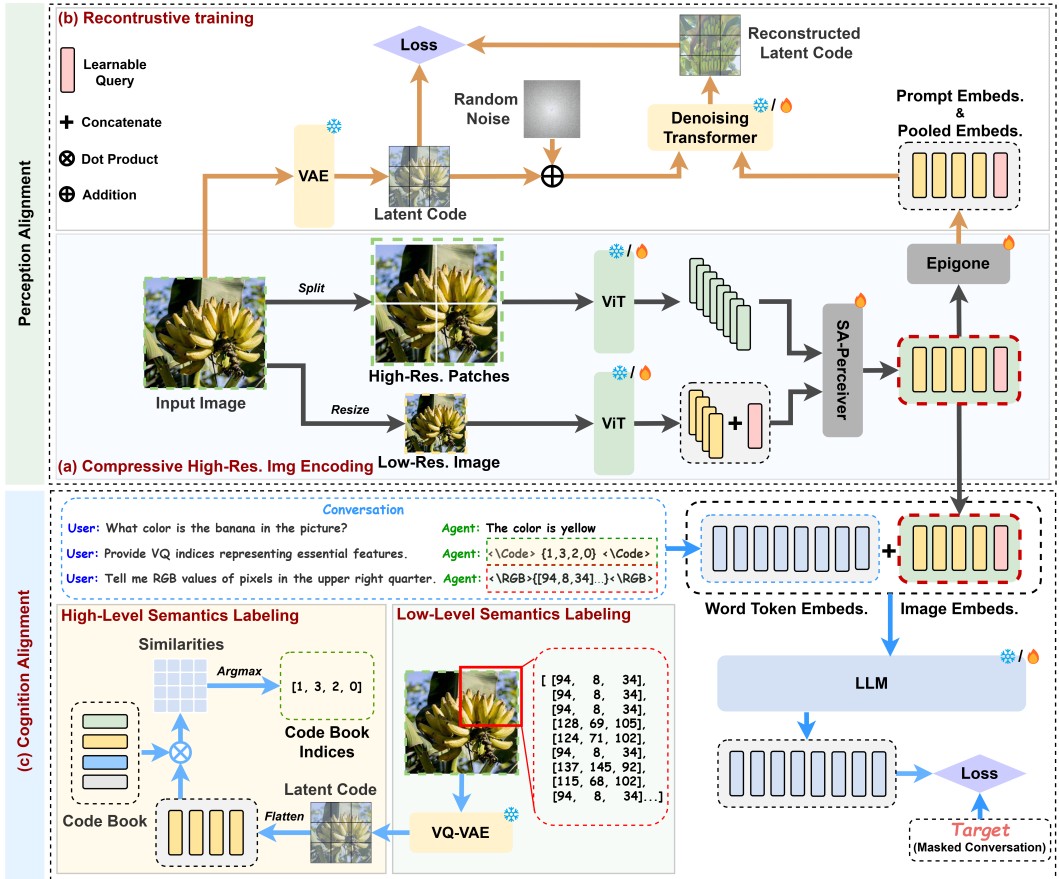

Figure 1: The Arichtecture of VLSA. TOP: Illustration of our perception alignment. (a) Compressive high-resolution image encoding and (b) reconstructive training concentrate high-resolution images into compact representations while reducing the information loss. Bottom: Illustration of our cognition alignment. The framework is required to predict the high-level semantics described by codebook indices and low-level semantics represented by pixel values when generating responses.

textual representations. Besides, the concatenation-based integration also involves fewer additional learnable parameters compared to the former method.

The feasibility of concatenation-based integration relies on the assumption that visual features have been well-aligned with textual features, enabling pretrained LLMs to understand visual inputs. Therefore, the alignment between vision and text determines **the lower bound** of MLLMs' performance, and improving this alignment through model architectures, training methods, and datasets is essential to enhance the versatility and reliability of MLLMs. It's important to note that this alignment focuses on mapping visual representations into the linguistic latent space, aiming for a distributional rather than a semantic alignment since vision inherently contains rich semantic information that is challenging to convey through text. Striving for a semantic-level transfer would result in considerable information loss and potentially compromise the performance of MLLMs in visual tasks. Existing approaches like LLaVA Liu et al. (2023b), PaLI Chen et al. (2022), and CogVLM Wang et al. (2024) utilize a linear layer or MLP as the projector to bridge the gap between visual and linguistic features, while models such as BLIP2 Li et al. (2023d), InstructBLIP Dai et al. (2023), and Qwen-VL Bai et al. (2023) leverage Q-former (also named perceiver) for feature alignment. Based on cross-attention, the Q-former transfers arbitrary visual sequences into a fixed-length query. However, both methods have limitations. The efficiency of the projector is compromised in scenarios involving the management of high-resolution image patches or the simultaneous processing of multiple images, attributable to the extended length of visual sequences. Furthermore, its dependency on the LLM decoder to discern the interrelations among visual contexts is notable. However, the causal attention mechanism inherent in LLMs exhibits limitations in accurately modeling visual embeddings' interrelationships. Some recent research (e.g., Xie et al. (2024); Zhou et al. (2024)) has demonstrated significant performance

improvements by enabling bi-directional attention for visual tokens in LLMs. (Nevertheless, this modification significantly alters the behavior of LLMs during reasoning, leading to increased demand for training data.) On the other hand, the Q-former is a lossy compressor that may overlook essential spatial and low-level features, even if it shortens visual sequences for improved efficiency.

Since the projector and the Q-former primarily align VL features at the distributional rather than the semantic level. The entire system heavily depends on the solid decoding capability of LLMs to convert sub-optimally aligned visual features into the desired word sequences. Therefore, we hypothesize that the alignment of LLMs' cognition with visual semantics determines **the upper bound** of MLLMs' performance. This perspective is reinforced by current techniques that prioritize substantial updates to LLMs' pretrained weights rather than freezing them. However, existing techniques focus exclusively on visual instruction tuning without explicit constraints for achieving semantic-level alignment. This limitation in current training paradigms led to a greater need for high-quality, large-scale VL datasets. Also, this demand becomes even more pronounced when dealing with the longer visual sequences introduced from high-resolution and multi-view contexts.

In response to these limitations, we proposed two design principles to improve MLLMs further. 1) To achieve better VL representation alignment, the image encoder and the VL alignment layer (e.g., MLP or Q-former) should generate concise visual sequences while minimizing information loss. Additionally, the VL alignment layer should incorporate a modeling approach (e.g., spatial modeling) that extends beyond causality among visual contexts. 2) To enhance the alignment of LLM cognition with vision, the LLM should possess a comprehensive understanding of visual semantics and should be directed to generate content based on its visual comprehension rather than relying solely on linguistic knowledge. We also introduce VL superior alignment (VLSA) to implement these principles. Specifically, VLSA decouples the alignment of the LLM with visual inputs into two stages: **perception alignment** and **cognition alignment**. In perception alignment, we first design the compressive high-resolution image encoding to concentrate information from high-resolution patches into a down-sampled version of the image using a cross-attention-based module called SA-perceiver. This process significantly reduces the image sequence length while preserving the spatial structure of the image. After that, we propose reconstructive training, which draws inspiration from the denoising process of Latent Diffusion Models (LDM) (e.g., Stable Diffusion Esser et al. (2024)), enabling the image encoder to capture more details and helps the SA-perceiver to be a lossless compressor as much as possible by reconstructing input images from random noise, using image embeddings as clues. In cognition alignment, we propose innovative explicit cognition alignment training tasks to facilitate the LLM in recognizing both high-level and shallow (i.e., pixel-level) visual semantics. In particular, we utilize a frozen VQ-VAE to discretely encode images and utilize its codebook indices as the tokens of high-level visual semantics. The MLLM is then required to predict the codebook indices and pixel values of certain areas in images. To summarize:

- We rethinking the modality alignment in MLLMs and propose VLSA, which consists of novel perception alignment and cognition alignment, to facilitate the incorporation of visual information in the LLM's reasoning process.

- We present compressive high-resolution image encoding and reconstructive training with LDM to prevent loss of information in visual encoding while decreasing the computational overhead during model inference. We also propose cognition alignment training tasks to enhance MLLMs' holistic comprehension of visual inputs by simultaneously predicting high-level and low-level image semantics.

- Comprehensive experimental evaluations across 20 benchmarks, accompanied by rigorously constructed ablation studies, underscore the efficacy and essentiality of designs in VLSA.

## 2 RELATED WORK

**Multi-modal Large Language Models (MLLMs).** Current advancements in Large Language Models (LLMs) Brown et al. (2020); Devlin et al. (2018); cha (2023); Zhang et al. (2022); Touvron et al. (2023); OpenAI (2023); Jiang et al. (2024) have demonstrated their growing reasoning capabilities and expansive knowledge base that are remarkably superior to traditional techniques. Consequently, there is an increasing trend of leveraging these readily available language-only models as the pivot in addressing multi-modal challenges. Flamingo Alayrac et al. (2022) and BLIP2 Li et al. (2023d) are pioneers in this field. Flamingo incorporates visual features by adding extra cross-attention layers

to every LLM layer, while BLIP2 uses Q-Former, a cross-attention-based module, to perform VL alignment. This module is trained using contrastive learning and generative tasks and compresses image information into a shorter form in linguistic feature space. The aligned image features are treated as the regular textual inputs fed to LLMs. Recently, LLaVA Liu et al. (2023b) simplifies the architecture of BLIP2 and adopts a concise linear projector rather than the Q-Former to facilitate image-text space alignment with minimal learnable parameters. Building upon these advancements, subsequent research endeavors are focused on improving the multi-modal capacities of foundational MLLMs. As contributors in this field, our work aims to enhance the visual processing abilities of LLMs by addressing the limitations of current VL alignment methods and introducing novel perception alignment and cognition alignment approaches.

**Refined improvements for basic MLLMs.** The methodologies employed in recent research endeavors can be broadly classified into four distinct groups. **(A)** Zhu et al. (2023); Zhang et al. (2023c); Zhao et al. (2023); Li et al. (2023b); Dai et al. (2023) further improve VL performances by fine-tuning models on enriched high-quality visual instruction-following datasets. These include, but are not limited to, instruction-following data that demand fine-grained visual inquiries or span various VL tasks, thereby pushing the frontier of MLLM capabilities further. **(B)** Liu et al. (2024a); Li et al. (2023a; 2024); Bai et al. (2023); Zhang et al. (2023a); Xu et al. (2024); Young et al. (2024) aid in improving VL performances by enhancing the resolution of encoded images or using more powerful vision encoders to offer richer visual information in VL reasoning and minimize hallucinations. However, dealing with high-resolution images significantly increases the input sequence length of LLMs and imposes a substantial computational burden. Therefore, these techniques also explore the effective compression of visual features to compromise the contradiction between information loss and limited computational resources. **(C)** Luo et al. (2024); Li et al. (2023f); Lin et al. (2024); Jiang et al. (2024) employ a mixture-of-experts (MoE) approach in VL alignment or LLMs' reasoning. They introduce duplicated visual projectors or feed-forward modules initialized with identical weights for MLLMs. This approach helps bridge the domain gap between vision and language and accommodates the distinction between following single-modality and multi-modal instructions. These techniques increase the model's capacity while maintaining a consistent inference cost by only activating a limited proportion of total parameters. **(D)** Wang et al. (2024); Gao et al. (2023); Zhang et al. (2023b) avoid the shallow VL late fusion, which entails concatenating VL representations as a single sequence to be fed into the first layer of LLMs. Instead, they explore the deep fusion of VL in each self-attention layer of LLMs. As the result of introducing additional visual experts or trainable prefixes to integrate visual representations into the hidden states of LLMs, they facilitate the MLLM's understanding of visual inputs.

By summing up the above works, we believe that the ensuring of strong alignment is one of the main factors contributing to the potential of MLLMs. Specifically, (B) enhances the alignment of VL representations, (D) facilitates the alignment of LLMs' understanding with vision, (A) and (C) benefit both types of alignment. Our approach focuses on facilitating alignment from perspectives similar to (B) and (D). However, we distinctively introduce explicit constraints, i.e., reconstructive loss and explicit cognition alignment, into both types of alignment instead of relying solely on end-to-end training with large-scale data to acquire these traits. Additionally, we minimize modifications to the LLM's structure. As a result, our approach offers the potential for better data efficiency.

## 3 METHODOLOGY

### 3.1 PERCEPTION ALIGNMENT

**Compressive High-resolution Image Encoding.** As shown in Fig 1 (A), our framework takes an image $X$ with arbitrary height and width as input. The initial step involves padding $X$'s dimension to $\mathbb{R}^{3 \times H \times W}$, which is the smallest dimension that can be evenly divided by $(h, w)$, the input size of the image encoder. Following this, $X$ is split into high-resolution patches $X_{\text{Hi}} \in \mathbb{R}^{m \times 3 \times h \times w}$, with $m$ representing the number of patches. Simultaneously, $X$ is down-sampled to create a low-resolution snapshot $X_{\text{Lo}} \in \mathbb{R}^{3 \times h \times w}$. Both $X_{\text{Hi}}$ and $X_{\text{Lo}}$ are then separately encoded by a vision encoder (e.g., CLIP Radford et al. (2021)), resulting in visual embedding $V_{\text{Hi}} \in \mathbb{R}^{m \times l \times d}$ and $V_{\text{Lo}} \in \mathbb{R}^{l \times d}$, where $l$ is the sequence length and $d$ is the feature dimension. Deviating from previous techniques that involve flattening visual embeddings into a long sequence for feed into the LLM, we gather information from $V_{\text{Hi}}$ while preserving their spatial structure according to $V_{\text{Lo}}$. This process is achieved through the SA-Perceiver depicted in Fig 2.

Specifically, we append a learnable query $q \in \mathbb{R}^{1 \times d}$ to $V_{\text{Lo}}$ in order to gather global features of the image, resulting $V'_{\text{Lo}} \in \mathbb{R}^{(l+1) \times d}$. Subsequently, we update $V'_{\text{Lo}}$ with $V_{\text{Hi}}$:

$$Q = w_{\text{q}} V'_{\text{Lo}}, \quad K = w_{\text{kv}} V_{\text{Hi}}, \quad V = w_{\text{kv}} V_{\text{Hi}},$$
$$V'_{\text{Lo}} = w_{\text{o1}} \left( \sigma \left( Q \times K \right) V \right) + V'_{\text{Lo}}, \tag{1}$$

where $w_{\text{q}}$, $w_{\text{kv}}$, $w_{\text{o1}}$ with the dimension of $\mathbb{R}^{d \times d}$ are linear projectors, $\sigma$ is the activation function SiLU Elfwing et al. (2018). After that, we utilize a self-attention process on $V'_{\text{Lo}}$ to allow the aggregated information from $V_{\text{Hi}}$ to pass among different image regions and facilitate the global feature extraction:

$$V'_{\text{Lo}} = w_{\text{o2}} \left( \left( V'_{\text{Lo}} \times w_{\text{k}} V'_{\text{Lo}} \right) + V'_{\text{Lo}} \right) V'_{\text{Lo}}. \tag{2}$$

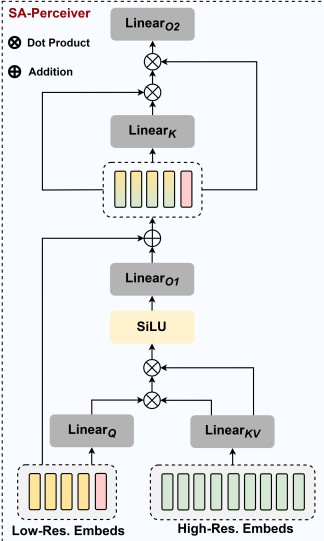

Figure 2: The details of the SA-Perceiver

Finally, we split $V'_{\text{Lo}}$ into image embeddings $V \in \mathbb{R}^{l \times d}$ and global embedding $P \in \mathbb{R}^d$ for the subsequent reconstructive training. Compared with general routines, our approach alleviates the challenge of modeling visual content solely based on causal attention in LLMs, while also reducing the computational overhead in the LLMs' inference process. These merits contribute to MLLMs in improving their ability to comprehend high-resolution input.

**Reconstructive training with LDM.** We propose a reconstructive constraint to restore $X$ from visual embeddings to minimize information loss in MLLMs' perception. Taking inspiration from Latent Diffusion Models (LDM), in practice, we require the model to be able to reconstruct $X$ from Gaussian noise under the condition of visual embeddings. As illustrated in Fig 1 (b), the process begins with encoding $X$ into the latent code $z \in \mathbb{R}^{s \times d}$ through a frozen Variational Auto-Encoder (VAE) Kingma (2013), where $s$ and $d$ represent the length and dimension, respectively. $z$ will be served as the target for reconstructive training. Following this, we introduce Gaussian noise to $z$, the intensity of which depends on the time step $t$, yielding a noisy latent $z_t$ at a random time step $t$. The crux of reconstructive training lies in optimizing a denoising transformer that reconstructs $z$ from $z_t$ using $V$ and $P$ as guidance. Both VAE and the denoising transformer are initialized from the open-sourced text-to-image generation model Stable Diffusion 3-medium (SD) Esser et al. (2024). The loss of this process can be formulated as:

$$\mathcal{L}_{\text{rec}} = \| z - z_\theta \left( z_t, \ c \right) \|_2^2, \tag{3}$$

where $z_\theta$ is the denoised latent predicted by the denoising transformer with learnable parameter $\theta$, $c \in \mathbb{R}^{(l+1) \times d}$ represents the combination of prompt embeddings and pooled embeddings carrying linguistic semantics required by SD. $c$ is defined as:

$$c = Epigone \left( V, \ P \right). \tag{4}$$

Here, the Epigone is a projection module that translates visual embeddings to prompt embeddings, in other words, generating pseudo image caption embeddings that describe the whole image. Deviating from the word embedding layers in the original SD, the Epigone is designed to offer detailed image information rather than solely high-level semantics that can be expressed using "real" words. As a result, it reduces randomness in reconstruction. To ensure computational efficiency, we employ a simple MLP projector to realize Epigone in this article. However, a more sophisticated module design may potentially lead to better performance. Moreover, there are two main reasons for using LDMs rather than conventional Auto-Encoders (AE) for reconstructive training: (1) Employing pretrained text-to-image LDMs can reduce information loss during image encoding while also promoting the alignment of visual and textual representations. (2) Pretrained text-to-image LDMs excel at accomplishing reconstructive tasks from a semantic perspective, thereby aiding in extracting rich visual semantics during image encoding.

## 3.2 COGNITION ALIGNMENT

We come up with the concept of cognition alignment that explicitly strengthens MLLMs in understanding high-level visual semantics and paying more attention to low-level image appearances. To discretely label images' high-level semantics, we propose leveraging the codebook indices of

VQ-VAE (Vector Quantized-Variational AutoEncoder) van den Oord et al. (2018) as targets. We first generate the latent code $L \in \mathbb{R}^{h' \times w' \times d'}$ of $X$ via VQ-VAE's encoder (initialized from Tang et al. (2023)) then calculate the similarity between $L$ and vector embeddings $B \in \mathbb{R}^{s \times d'}$ in the codebook. The corresponding codebook indices $Target_{\text{VQ}} \in \mathbb{R}^{h'w'}$ are formalized as:

$$Target_{\text{VQ}} = \text{argmin} \left( B \times \text{flatten} \left( L \right) \right). \tag{5}$$

On the other hand, the labels of low-level image appearances are represented by pixels' RGB values. Upon receiving the coordinates of a random vertex, the model is tasked with predicting $Target_{\text{PX}}$, which comprises the RGB values within a quarter of the area of $X$ with this vertex as the top-left corner. We process $Target_{\text{VQ}}$ and $Target_{\text{PX}}$ and incorporate them into instruction following datasets (e.g., Alpaca Taori et al. (2023), LLaVA-665k Liu et al. (2023a)) using the provided templates in the appendix. Especially, we introduce random positions in conversations for inserting queries to predict codebook indices and pixel values and require Language Models to generate $Target_{\text{VQ}}$ and $Target_{\text{PX}}$ based on both visual contexts $V$ and textual contexts, which is denoted as $I$. In this way, we facilitate visual reasoning based on semantic predictions. Consequently, various downstream tasks can benefit from our cognition alignment.

## 3.3 Response Generation

The response generation of the LLM commences with encoding an arbitrary user instruction $I$ into word token embeddings $I_{\text{emb}} \in \mathbb{R}^{i \times d}$. Subsequently, $I_{\text{emb}}$ is amalgamated with image embeddings $V$, and they are collectively inputted into the LLM to generate the desired response $R$, which is formulated as:

$$R = g_{\text{LLM}} \left( \text{concat} \left[ V, \ I_{emb} \right] \right), \tag{6}$$

where $g_{\text{LLM}}$ symbolizes the reasoning process of the LLM and $\text{concat} \left[ \cdot \right]$ denotes the concatenation of features. During training, the optimization objective can be formulated by:

$$\text{argmin} \, \mathcal{L} \left( R, Target; \ \theta_{\text{LLM}} \right). \tag{7}$$

Here, $Target$ refers to the augmented ground-truth response that contains $Target_{\text{VQ}}$ and $Target_{\text{PX}}$ for cognition alignment, $\mathcal{L} \left( \cdot \right)$ denotes the objective loss function, and $\theta_{\text{LLM}}$ represents the parameters of the LLM. $\mathcal{L} \left( \cdot \right)$ is defined as:

$$\mathcal{L} = \sum_{i=1}^{B} \sum_{j=1}^{K+1} \log p \left( y_j^i | V^i, \ I_{emb}^i, \ Target_{0:j-1}^i; \ \theta_{\text{LLM}} \right), \tag{8}$$

where $B$ denotes the batch size, and $K$ is the length of the response $R$. The final loss for the entire system is determined through the summation of $\mathcal{L}$ and $\mathcal{L}_{\text{rec}}$.

## 4 Experiments

### 4.1 Implementation Details

To assess the capabilities of VLSA, we integrate it into the prevalent MLLM architecture LLaVA-Next (LLaVA-1.6) Dubey et al. (2024). This architecture simply adopts a MLP as the projector to connect the vision encoder and the language model. The conciseness of this setup enables us to analyze the impact of each component in VLSA better. For the LLM backbone, we apply the LLaMA3-8B-Instruct model Dubey et al. (2024), a fine-tuned version of LLaMA3-8B that improves instruction following. It has 32 transformer layers with feature dimension $d = 4096$. For the vision encoder, we apply CLIP-ViT-L/14@336px Radford et al. (2021), which has 24 transformer layers with feature dimension $d = 1024$. The low-resolution snapshot $X_{\text{Lo}}$ in Sec 3.1 has dimensions of $\mathbb{R}^{3 \times 336 \times 336}$, which matches the size of the input images for the vision encoder. The length of the image embedding $V$ input to the LLM remains consistently at 576, representing a reduction of up to **four-fold** compared to the original LLaVA-Next. Epigone in Equation 4 is a low-rank MLP, which consists of two linear layers and the SiLU activation function. Its input and output dimensions are both 4086, while the intermediate dimension is 256. Our development of VLSA is based on the original LLaVA-Next codebase with minor modifications. During the inference, We use greedy decoding with a temperature of 0 to ensure reproducibility.

Table 1: **Comparisons on academic-task-oriented datasets**. *denotes a larger actual receptive field. [†]Includes in-house data that is not publicly accessible. Res, PT, and IT indicate input image resolution and the number of samples in the pretraining and the finetuning, respectively. Benchmark names are abbreviated due to space limits. VQA[v2] Goyal et al. (2017); GQA Hudson & Manning (2019); VisWiz Gurari et al. (2018); TextC: TextCaps Sidorov et al. (2020); COCO Chen et al. (2015); VQA[ST]: ST-VQA Biten et al. (2019); SQA[I]: ScienceQA-IMG Lu et al. (2022); SQA: ScienceQA Lu et al. (2022);VQA[T]: TextVQA Singh et al. (2019).

| Method | LLM | Res. | PT | IT | VQA[v2] | GQA | VisWiz | COCO | TextC | VQA[ST] | SQA[I] | SQA | VQA[T] |
|---|---|---|---|---|---|---|---|---|---|---|---|---|---|
| BLIP-2 | Vicuna-13B | 224 | 129M | - | 41.0 | 41 | 19.6 | – | – | 36.4 | 61 | - | 42.5 |
| InstructBLIP | Vicuna-7B | 224 | 129M | 1.2M | – | 49.2 | 34.5 | 102.1 | 97.9 | 38.1 | 60.5 | - | 50.1 |
| InstructBLIP | Vicuna-13B | 224 | 129M | 1.2M | – | 49.5 | 33.4 | – | – | 38.7 | 63.1 | - | 50.7 |
| Shikra | Vicuna-13B | 224 | 600K | 5.5M | 77.4 | – | – | – | – | – | – | - | – |
| IDEFICS-9B | LLaMA-7B | 224 | 353M | 1M | 50.9 | 38.4 | 35.5 | – | 25.4 | – | – | - | 25.9 |
| IDEFICS-80B | LLaMA-65B | 224 | 353M | 1M | 60.0 | 45.2 | 36.0 | – | 56.8 | – | – | - | 30.9 |
| Qwen-VL | Qwen-7B | 448 | 1.4B[†] | 50M[†] | 78.8 | 59.3 | 35.2 | – | – | – | 67.1 | - | 63.8 |
| Qwen-VL-Chat | Qwen-7B | 448 | 1.4B[†] | 50M[†] | 78.2 | 57.5 | 38.9 | – | – | – | 68.2 | - | 61.5 |
| LLaVA-1.5 | Vicuna-7B | 336 | 558K | 665K | 76.6 | 62.0 | 50.0 | 109.4 | 101.8 | 54.0 | 69.8 | 70.0 | 58.2 |
| LLaVA-Next | LLaMA3-8B | Any | 558K | 790K | 82.4 | 64.9 | 46.7 | 137.3 | 70.1 | 64.2 | 74.6 | 72.1 | 63.9 |
| LLaVA-Next | LLaMA3-8B | Any | 980K | 790K | 83.6 | 64.6 | 49.9 | 136.4 | 68.3 | 65.0 | 75.1 | 74.2 | 64.8 |
| Δ | - | - | - | - | +1.2 | -0.3 | +3.2 | -0.9 | -1.8 | +0.8 | +0.5 | +2.1 | +0.9 |
| VLSA | LLaMA3-8B | 336* | 980K | 790K | 83.3 | 65.3 | 57.7 | 139.5 | 73.3 | 65.7 | 77.5 | 78.6 | 65.2 |
| Δ | - | - | - | - | +0.9 | +0.4 | +11.0 | +2.2 | +3.2 | +1.5 | +2.9 | +6.5 | +1.3 |

**Training strategy.** The training of VLSA consists of three stages. In Stage I, the focus is on the optimization of the SA-perceiver and the Epigone module, while maintaining the constancy of other parameters. The dataset employed for this initial phase aligns with the 558K instances previously utilized in the pre-training of LLaVA-Next. Transitioning to Stage II, we further enable the updating of both the denoising transformer and the vision encoder. Given the intrinsic limitations of diffusion models in generating small text in the picture, there exists a risk of the framework neglecting nuanced details (such as tiny textual elements) after the reconstructive training. To counterbalance this shortcoming, the framework undergoes training utilizing the LLaVAR Zhang et al. (2023c) dataset, comprising 422K instructions explicitly designed to bolster OCR capabilities within text-rich images. Finally, Stage III entails a comprehensive optimization of all parameters, utilizing a fine-tuning dataset containing 790K instructions as leveraged by LLaVA-Next. To facilitate fairness comparisons, the performance of LLaVA-Next pre-trained on 980K instances, including LLaVAR, is also reported. More details are provided in the appendix.

**Competitive methods.** We consider BLIP-2 Li et al. (2023d), InstructBLIP Dai et al. (2023), Shikra Chen et al. (2023), IDEFICS Laurençon et al. (2023), Qwen-VL Bai et al. (2023), LLaVA-1.5 Liu et al. (2023a) and primarily focus on comparisons with the baseline model LLaVA-Next.

## 4.2 EXPERIMENTAL RESULTS

**Quantitative Results.** We first compare VLSA with competitive methods on 8 academic-task-oriented datasets (each examines a specific ability), including 6 VQA tasks: VQA[v2] Goyal et al. (2017), GQA Hudson & Manning (2019), VisWiz Gurari et al. (2018), ST-VQA Biten et al. (2019), ScienceQA-IMG Lu et al. (2022), TextVQA Singh et al. (2019) and 2 image captioning tasks: TextCaps Sidorov et al. (2020), COCO Chen et al. (2015). As shown in Tab 1, VLSA demonstrates a significant and consistent improvement compared to the baseline LLaVA-Next. Specifically, VLSA notably enhances performance on previously unseen datasets VisWiz (+11.0) and ScienceQA (+6.5). However, VLSA does not lead to a notable performance boost on VQA[v2], GQA, and COCO. This could be due to the fact that images or annotations from these datasets were observed during the fine-tuning (the 790K fine-tuning set from LLaVA-Next includes their training sets). As a result, the baseline model also performs well on these datasets.

Besides, we further assess VLSA's multi-modal comprehension ability on 7 instruction-following benchmarks that are tailor-made for MLLMs, including POPE Li et al. (2023e), MME Fu et al. (2023), MMBench Liu et al. (2023c), MMBench-Chinese Liu et al. (2023c), SEED-Bench Li et al. (2023c), LLaVA-Bench (In-the-Wild) Liu et al. (2023b) and MM-Vet Yu et al. (2023). As shown in Tab 2, VLSA achieves obviously better results compared to previous generalist models on MME (+91), SEED (+5.6), and MM-Vet (+6.3). The only performance degradation occurs in MMB (-0.3). However, we notice that the baseline model, which includes an additional 442K OCR instances during pre-training, shows a more significant performance drop (-2.5). Therefore, we believe that the anomaly in MMB is due to the challenge of optimizing for both OCR instructions and other

Table 2: **Comparison for multi-modal comprehension on MLLM benchmarks**. POPE Li et al. (2023e); MME Fu et al. (2023); MMB: MMBench Liu et al. (2023c); MMB$^{CN}$: MMBench-Chinese Liu et al. (2023c); SEED: SEED-Bench Li et al. (2023c); LLaVA$^W$: LLaVA-Bench (In-the-Wild) Liu et al. (2023b); MM-Vet Yu et al. (2023). [†]Includes in-house data that is not publicly accessible. [‡]Evaluating via text-only GPT-4-0613. [*]denotes a larger actual receptive field.

| Method | LLM | Res. | PT | IT | POPE | MME | MMB | MMB$^{CN}$ | SEED | LLaVA$^W$ | MM-Vet |
|--------|-----|------|----|----|------|-----|-----|------------|------|-----------|--------|
| BLIP-2 | Vicuna-13B | 224 | 129M | - | 85.3 | 1293.8 | – | – | 46.4 | 38.1[‡] | 22.4[‡] |
| InstructBLIP | Vicuna-7B | 224 | 129M | 1.2M | – | – | 36 | 23.7 | 53.4 | 60.9[‡] | 26.2[‡] |
| InstructBLIP | Vicuna-13B | 224 | 129M | 1.2M | 78.9 | 1212.8 | – | – | – | 58.2[‡] | 25.6[‡] |
| Shikra | Vicuna-13B | 224 | 600K | 5.5M | – | – | 58.8 | – | – | – | – |
| IDEFICS-9B | LLaMA-7B | 224 | 353M | 1M | – | – | 48.2 | 25.2 | – | – | – |
| IDEFICS-80B | LLaMA-65B | 224 | 353M | 1M | – | – | 54.5 | 38.1 | – | – | – |
| Qwen-VL | Qwen-7B | 448 | 1.4B[†] | 50M[†] | – | – | 38.2 | 7.4 | 56.3 | – | – |
| Qwen-VL-Chat | Qwen-7B | 448 | 1.4B[†] | 50M[†] | – | 1487.5 | 60.6 | 56.7 | 58.2 | – | – |
| LLaVA-1.5 | Vicuna-7B | 336 | 558K | 665K | 85.9 | 1462.6 | 64.8 | 57.6 | 58.6 | 63.2[‡] | 30.6[‡] |
| LLaVA-Next | LLaMA3-8B | Any | 558K | 790K | 87.7 | 1753 | **72.2** | 70.1 | 60.0 | 69.4[‡] | 33.4[‡] |
| LLaVA-Next | LLaMA3-8B | Any | 980K | 790K | 88.0 | 1776 | 69.7 | 70.3 | 60.4 | 69.8[‡] | 36.1[‡] |
| Δ | - | - | - | - | +0.3 | +23 | -2.5 | +0.2 | +0.4 | +0.4 | +2.7 |
| VLSA | LLaMA3-8B | 336* | 980K | 790K | **88.6** | **1844** | 71.9 | **71.5** | 65.6 | 72.1[‡] | 39.7[‡] |
| Δ | - | - | - | - | +0.9 | +91 | -0.3 | +1.4 | +5.6 | +2.7 | +6.3 |

general instructions simultaneously. **Note** that there are discrepancies between the results of our reproduced LLaVA-Next and those reported officially. This is because we used the CLIP vision encoder instead of the SigLIP vision encoder. Additionally, constrained by computational resources, we are compelled to utilize a smaller equivalent batch size.

**Qualitative Results.** To underscore the efficacy of our reconstructive training, we conduct an analytical visualization comparing the visual features extracted by the CLIP encoder, inclusive of projector layers, in scenarios with and without the application of reconstructive training. As shown in Fig 3, the CLIP encoder fine-tuned solely through visual instruction following tasks tends to lose substantial semantics in the original images. This phenomenon may be associated with CLIP's inherent contrastive pretraining, as contrastive learning only focuses on high-level semantic consistency. By contrast, the CLIP encoder fine-tuned via reconstructive training can minimize the loss of detailed information, thus facilitating the alignment of MLLMs' perception with input images. To further demonstrate the advantages of perception alignment, we provide additional results on visual writing tasks in the Appendix.

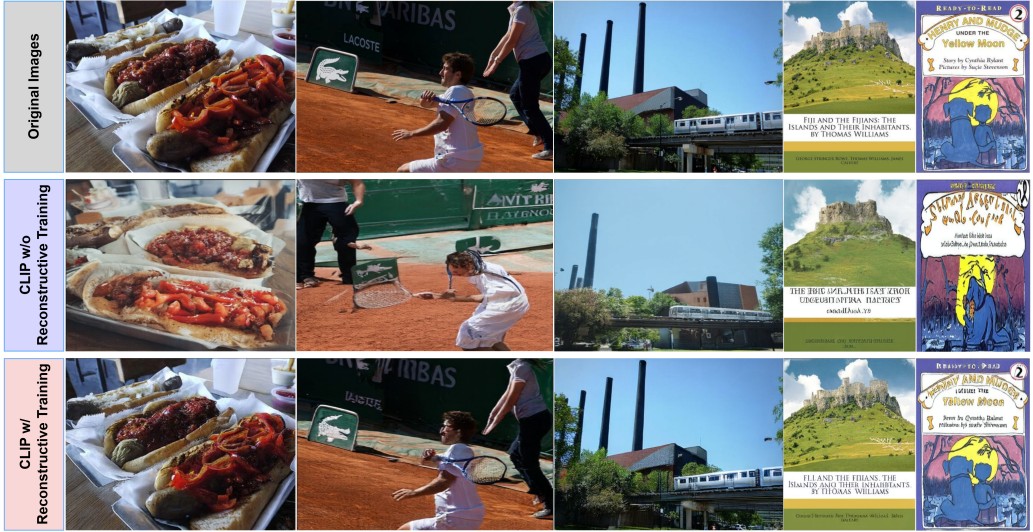

Figure 3: Qualitative demonstration of the reconstructive training, which significantly reduces information loss in the CLIP encoder.

## 4.3 ABLATION STUDY

We delve into an exhaustive exploration of the impact of components within VLSA, including (a) compressive high-resolution image encoding, (b) reconstructive training, and (c) cognition alignment.

Table 3: Ablation studies and results on documents understanding. AI2D Kembhavi et al. (2016); ChartQA Masry et al. (2022); DocVQA Mathew et al. (2021); MME$^C$:MME-Cognition Fu et al. (2023); MME$^P$:MME-Perception Fu et al. (2023); SQA$^I$: ScienceQA-IMG Lu et al. (2022); SQA: ScienceQA Lu et al. (2022); VisWiz Gurari et al. (2018); $^*$denotes a larger actual receptive field.

| Method | Res. | AI2D | ChartQA | DocVQA | MME$^C$ | MME$^P$ | SQA$^I$ | SQA | VisWiz |
|---|---|---|---|---|---|---|---|---|---|
| (1) LLaVA-Next | Any | 69.5 | 67.1 | 73.7 | 298.9 | 1455.7 | 72.1 | 74.6 | 46.7 |
| *Ablations on the Low Resolution Setting* | | | | | | | | | |
| (2) LLaVA-Next | 336 | 67.8 (-1.7) | 44.8 (-22.3) | 45.4 (-28.3) | 322.1 (+23.2) | 1417.3 (-38.4) | 76.0 (+3.9) | 78.3 (+3.7) | 48.9 (+2.2) |
| *Ablations on the Perception Alignment* | | | | | | | | | |
| (3) LLaVA-Next + (a) | 336$^*$ | 69.7 (+0.2) | 65.3 (-1.8) | 61.8 (-11.9) | 315.0 (+7.1) | 1426.5 (-29.2) | 76.3 (+4.2) | 78.1 (+3.5) | 55.9 (+9.2) |
| (4) LLaVA-Next + (ab) | 336$^*$ | 68.2 (-1.3) | 67.4 (+0.3) | **75.5** (+1.8) | 329.9 (+40.0) | 1481.4 (+25.7) | 74.1 (+2.0) | 76.8 (+2.2) | 53.6 (+6.9) |
| *Ablations on the Cognition Alignment* | | | | | | | | | |
| (5) LLaVA-Next + (c) | Any | **72.5** (+3.0) | 67.0 (-0.1) | 73.9 (+0.2) | **353.6** (+54.7) | **1570.0**(+114.3) | 72.6 (+0.5) | 75.5 (+0.8) | 50.9 (+4.2) |
| (6) LLaVA-Next + (abc) | 336$^*$ | 71.4 (+1.9) | **67.9** (+0.8) | 75.2 (+1.5) | 336.4 (+37.5) | 1507.4 (+51.3) | **77.5** (+4.2) | **78.6** (+5.4) | **57.7** (+11.0) |

To this end, we compare the results of six variants across eight benchmarks. Variants are (1) the original LLaVA-Next serving as the baseline, (2) directly reducing input resolution to mitigate computational costs, (3) LLaVA-Next incorporating (a), (4) LLaVA-Next integrating both (a) and (b), achieving our proposed perception alignment, and (5) LLaVA-Next solely employing (c). The eight datasets, categorized by the abilities they assess, include ChartQA Masry et al. (2022), DocVQA Mathew et al. (2021), and MME-perception, which primarily evaluate perceptive capabilities; MME-cognition, focusing on cognitive abilities; and AI2D Kembhavi et al. (2016), SQA-I, SQA, and VizWiz, which comprehensively assess both perception and cognition.

**Effectiveness of Perception Alignment.** Results in Table 3 (1) and (2) indicate that reducing the input resolution severely impairs perceptual capabilities. However, reducing the resolution outperforms the baseline on MME-cognition, SQA, and VisWiz, revealing that current techniques do not effectively model high-resolution input. Through (2) and (3), it can be observed that module (a) significantly mitigates the negative impact on perceptive abilities while also shortening visual feature sequences. As previously mentioned, (a) aggregates high-resolution features prior to their input into the LLM, instead of relying solely on the LLM's causal modeling to process them. The impressive performance enhancements on MME-cognition, SQA, and VisWiz indicate that this approach can enhance the model's capability to comprehend high-resolution inputs. Upon comparing (4) and (3), it becomes evident that the inclusion of module (b) effectively offsets the information loss caused by (a), thus improving the model's perceptual abilities to a point where it exceeds the baseline model. Nevertheless, we have observed that while reconstructive training focuses on retaining more image details, it also limits the improvement of cognitive abilities brought by (a).

**Necessity of Cognition Alignment.** By comparing (6) with (4), we observe that our cognition alignment explicitly encourages the LLM to understand both shallow and deep semantics of visual features, enabling more effective utilization of the visually rich semantic features after perception alignment and reducing the negative effects of module (b) on cognitive abilities, leading to consistent performance improvements. Additionally, we conducted isolated testing on the efficacy of cognition alignment in (5), which resulted in significant performance improvements on AI2D, MME-cognition, MME-perception, and VisWiz, highlighting the universal value of cognition alignment.

## 5 DISCUSSION AND CONCLUSION

This paper investigates the visual-language modality alignment in MLLMs and proposes novel concepts of perception and cognition alignment. Perception alignment aims to minimize information loss during visual encoding, while cognition alignment helps MLLMs comprehensively understand visual embeddings. The paper also introduces VLSA, which involves compressive high-resolution image encoding and reconstructive training to achieve perception alignment while reducing computational overhead, and training tasks that require MLLMs to simultaneously predict high-level and low-level image semantics, labeled by codebook indices of a pretrained VQ-VAE and RGB pixel values, to realize cognition alignment. Extensive experiments on various benchmarks consistently demonstrate that VLSA enhances performance. **Limitations & Future Work.** Considering the dual optimization objectives of VLSA, which include the reconstruction loss and LLM's autoregressive loss, solving the combinatorial optimization problem poses a significant challenge. The current approach involves simultaneously leveraging these two losses in all training phases without balancing their proportions, which may not be optimal. It could be beneficial to explore a reasonable loss ratio and strategies for their application at different training stages to improve the performance of VLSA further.

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
