# OpenReview forum: "Rethinking Modality Alignment in Multi-Modal Large Language Models"
_ICLR.cc/2025/Conference — Submitted to ICLR 2025_

### Official Review · Reviewer_Gehn · 2024-10-27

**Soundness:** 3
**Presentation:** 3
**Contribution:** 3
**Rating:** 5
**Confidence:** 5

**Summary:**

This paper introduces VL Superior Alignment (VLSA), a framework to improve visual and linguistic alignment in Multimodal Large Language Models (MLLMs). VLSA addresses alignment issues with two stages: perception alignment to reduce visual information loss using high-resolution encoding and Latent Diffusion Models (LDM), and cognition alignment to enhance understanding of visual semantics and image details through specific training tasks. Evaluated on 20 benchmarks, VLSA consistently improves performance and demonstrates the framework's effectiveness.

**Strengths:**

1. Innovative Approach to Modality Alignment:The paper introduces a novel two-stage approach to align visual and linguistic modalities, which is a significant advancement over traditional methods that often treat alignment as a secondary consideration.
2. Reduction of Information Loss: The proposed compressive high-resolution image encoding and reconstructive training are innovative ways to preserve detailed visual information, which is crucial for high-fidelity multimodal interactions.
3. Enhanced Visual Reasoning: The paper By addressing the limitations of causal attention mechanisms in capturing visual relationships, VLSA potentially improves the model's ability to reason about visual content, leading to more accurate and reliable outputs.

**Weaknesses:**

1. The introduction of new components like the SA-Perceiver and reconstructive training may increase the model's complexity and computational requirements, which could be a barrier for adoption in resource-constrained environments.
2. The dual optimization objectives (reconstruction loss and autoregressive loss) might be challenging to balance, and the paper does not discuss how these are weighted or optimized together.
3. It's not fully addressed how VLSA scales with increasing model size or input complexity. There's a concern about whether the benefits would diminish as the model grows larger or the inputs become more complex.

**Questions:**

1. The VLSA framework presented in the paper has demonstrated performance improvements across various benchmarks. However, do these benchmarks encompass a diverse enough range of tasks to attest to the generalization capabilities of VLSA?


2. How does VLSA perform in terms of computational efficiency, especially when deployed in resource-limited environments?


3. Could you please explain why MLLMs based on VQ and codebook learning are called cognition alignment? Given the current development of MLLMs, such as SOTA models like Chameleon, EMU3, and LAViT, the performance of these VQ-based MLLMs is still lower than that of LLaVA-One-Vision and InternVL2. Why does your method improve the cognitive performance of the model after converting images to VQ?


4. VQ-based MLLMs generally experience a significant drop in OCR performance. Could you include more OCR benchmarks, such as TextVQA, OCRBench, AI2D, and ChartQA, in the results?

---

> ### Author Response · Authors · 2024-11-19
> **Response to the reviewer Gehn**
>
> We sincerely appreciate the constructive feedback provided by the reviewer, which has been invaluable in enhancing the quality of our paper!
>
> Over the past week, we conducted the necessary experiments and thoughtfully addressed each concern raised by the reviewer. Due to system limitations that affected the proper display of our updated Figures and some Tables, we have included our responses in the Supplementary Material. Please refer to the detailed replies in **Gehn.pdf**, which are located within the **Supplementary Material.zip**.

---

> > ### Author Response · Authors · 2024-11-23
> >
> > Dear Reviewer Gehn,
> >
> > Thank you again for the time and effort spent on your thorough review of our paper. Since the author-reviewer discussion deadline is fast approaching, we kindly ask for feedback on our responses. We would be happy to discuss more if there are still some open questions.
> >
> > Best Regards,
> >
> > Authors

---

### Official Review · Reviewer_q7Gw · 2024-11-01

**Soundness:** 3
**Presentation:** 3
**Contribution:** 2
**Rating:** 3
**Confidence:** 4

**Summary:**

The paper presents VL Superior Alignment (VLSA), a novel framework aimed at improving modality alignment in Multi-modal Large Language Models (MLLMs) by addressing the misalignment issues in vision-language (VL) tasks. VLSA decouples alignment into two stages: Perception Alignment and Cognition Alignment. Experimental results across 20 MLLM benchmarks demonstrate VLSA’s improved performance over prior models.

**Strengths:**

Overall, this paper presents an innovative approach to vision-language alignment:

- Two-Stage Alignment Framework: VLSA introduces a unique two-stage alignment strategy, dividing perception (image encoding) and cognition (semantic comprehension). This separation minimizes visual information loss, enhancing model interpretability.
- Compressive Image Encoding: The proposed encoding method preserves spatial relationships within images, enabling efficient processing of high-resolution data while lowering computational costs.
- Reconstructive Training: Inspired by Latent Diffusion Models, this technique enables the model to recover original visual details, significantly improving visual reasoning accuracy.

**Weaknesses:**

1. Computational Complexity: Balancing the reconstruction and language modeling losses simultaneously can be challenging, potentially hindering optimization efficiency.
2. Limited Scalability: Given the computational complexity and the involvement of a two-stage alignment process, it is difficult to trust that this method has good scalability—an essential aspect for current VLMs.
3. Limited Experiments: The evaluation is restricted to conventional benchmarks (single-image input with textual output). Since the paper claims to improve visual-semantic understanding, could it demonstrate performance on benchmarks like BLINK [1] that involve visual prompts? Moreover, the authors assert that this is a general approach—have they considered testing on other interleaved benchmarks, such as DEMON [2] ?

[1] BLINK: Multimodal Large Language Models Can See but Not Perceive

[2] Fine-tuning multimodal llms to follow zero-shot demonstrative instructions

**Questions:**

see weakness

---

> ### Author Response · Authors · 2024-11-19
> **Response to the reviewer q7Gw**
>
> We sincerely appreciate the constructive feedback provided by the reviewer, which has been invaluable in enhancing the quality of our paper!
>
> Over the past week, we conducted the necessary experiments and thoughtfully addressed each concern raised by the reviewer. Due to system limitations that affected the proper display of our updated Figures and some Tables, we have included our responses in the Supplementary Material. Please refer to the detailed replies in **q7Gw.pdf**, which are located within the **Supplementary Material.zip**.

---

> > ### Author Response · Authors · 2024-11-23
> >
> > Dear Reviewer q7Gw,
> >
> > Thank you again for the time and effort spent on your thorough review of our paper. Since the author-reviewer discussion deadline is fast approaching, we kindly ask for feedback on our responses. We would be happy to discuss more if there are still some open questions.
> >
> > Best Regards,
> >
> > Authors

---

### Official Review · Reviewer_1gd8 · 2024-11-02

**Soundness:** 1
**Presentation:** 2
**Contribution:** 2
**Rating:** 5
**Confidence:** 3

**Summary:**

This paper introduces a new method for aligning visual inputs in MLLMs. The authors argue that current approaches to visual alignment are lacking and they do not thoroughly project image embeddings into the text space. They propose a novel architecture, which includes new modules such as perception alignment and cognition alignment. The perception alignment module tries to fix the issue caused by causal attention on image embeddings as that is not the best way to perform attention over image embeddings. They also introduce a new reconstructive training procedure based on LDMs which aim to improve the alignment between the text embeddings and the projected image embeddings and two auxiliary tasks predicting codebook indices from VQ-VAE and predicting the RGB pixel values on an image using an LLM to improve semantic and low-level image understanding of MLLMs. The new architecture and training method results in impressive performance on standard multi-modal benchmarks and provides new interesting ways to think about multi-modal alignment.

**Strengths:**

1) The paper introduces novel and interesting ideas to improve multi-modal alignment such as using LDMs and VQ-VAE codebook indices to improve semantic alignment between text and image features. While a lot of the choices lack theoretical justification/empirical observations, the ideas themselves might be useful for other work in the field/future work.

2) The new architecture and training paradigm gets impressive results on common multi-modal benchmarks.

**Weaknesses:**

1) Several hypothesis lack theoretical/empirical justification: In the introduction and method section, the authors provide several hypothesis which aren't backed by theoretical justifications or empirical observations. Moreover, the introduction section is not very well written and it is hard to follow some of the claims. In the method section, while the section contains all technical details, the motivation behind some design choices is not very clear. For instance:

a) "the alignment between vision and text determines the lower bound of MLLMs’ performance" and "Therefore, we hypothesize that
the alignment of LLMs’ cognition with visual semantics determines the upper bound of MLLMs’ performance". It is not clear why these two statements are true. There are several other factors which influence the performance of MLLMs and it is unclear why only the alignment between vision and text or the semantic alignment will be deciding the lower and upper bounds. It will be helpful if the authors can provide some additional context/ some sources where these claims are made or some experiments which help better support these claims.

b) "It’s important to note that this alignment focuses on mapping visual representations into the linguistic latent space, aiming for a distributional rather than a semantic alignment since vision inherently contains rich semantic information that is
challenging to convey through text.": Could the authors please elaborate on what this means exactly? It will be good to have a more theoretical understanding of why the current alignment techniques are distributional rather than semantic. For example, looking at visualizations of the projections or looking at attention maps might help better support this claim.


2) Incomplete experiments and ablations: There are claims made in the paper which are not backed by experiments, and moreover a lot of design choices are not ablated well (it is unclear how the design choice was made). For instance:

a) The authors claim that the SA-Perceiver module leads to lower computational overhead during model inference, but no latency experiments/ theoretical analysis is provided as to why this is true. Given the fact that there is additional computation for computing the high resolution image features and the cross-attention with the low resolution image features, it is not clear why the architecture would be computationally more efficient during inference compared to a linear projector. It might be helpful if the authors can provide some latency analysis showing that inference using VLSA is faster compared to vanilla architectures. One easy way to show this could be to run inference over a fixed set of images and calculate the time taken per image for the different architectures and across different image resolutions.

b) The architecture of the SA-perceiver module lacks sufficient ablations as there aren't any ablations which compare using high resolution image features only, or other mechanisms or merging the high resolution features and low resolution features (for example concatenation followed by self-attention). Also having some analysis on how changing the lower resolution impacts performance would be interesting. Moreover, it is not clear why there needs to be a learnable parameter P in the low-resolution image features since that global embedding is not distinctly used anywhere else in the architecture. Adding an ablation with only high-resolution image features, high-resolution features + self-attention, and [high-resolution features,low-resolution features] + self-attention could be a useless analysis to conduct.

c) In the cognition alignment module, there are no ablations which compare using only the codebook task vs using the rgb task. Also, it would be helpful to have some more analysis on how exactly these tasks help other than just looking at the performance boosts on benchmarks. For example, having more results which show that training with codebook data is able to improve semantic understanding of the image, or showing that training with rgb value prediction leads to improved low-level image recognition might add more support to the architecture design choice.

d) All experiments and ablations are only done with the Llava-Next architecture and only LLama3-8B as the LLM, so it is unclear if these improvements in benchmarks also generalize to other MLLM architectures which can support the SA-Perceiver module. It would be good to have more experiments across different LLM architectures and sizes (Vicuna 7B, LLama 3B, etc) and also across different architectures (Shikra, Qwen2VL, Mini-GPT4) etc.



3) Architecture and training paradigm is quite complex: There are several additional components in the architecture and it uses a 3 phase training procedure, adding significant overhead in terms of number of additional modules needed (LDM, VQ-VAE etc) and also training time. While the performance gains on benchmarks is impressive, it also severely limits the generalizability and extensibility of this architecture. It might be interesting to look at the trade-off between the complexity and performance by simplifying the architecture a bit.



4) Minor nitpicks: In equation 5, the shape of of TargetVQ should be (sxh'xw') assuming the X operation refers to dot product.

**Questions:**

Please refer to the weaknesses section. Additionally:

1) How were the tasks for the cognition alignment module chosen? It would be good to have some insight into why the codebook and the rgb value prediction task were chosen.

2) Have you experimented with adding the image caption embeddings to the denoising transformer alongside the epigone embeddings?

---

> ### Author Response · Authors · 2024-11-19
> **Response to the reviewer 1gd8**
>
> We sincerely appreciate the constructive feedback provided by the reviewer, which has been invaluable in enhancing the quality of our paper!
>
> Over the past week, we conducted the necessary experiments and thoughtfully addressed each concern raised by the reviewer. Due to system limitations that affected the proper display of our updated Figures and some Tables, we have included our responses in the Supplementary Material. Please refer to the detailed replies in **1gd8.pdf**, which are located within the **Supplementary Material.zip**.

---

> > ### Author Response · Authors · 2024-11-23
> >
> > Dear Reviewer 1gd8,
> >
> > Thank you again for the time and effort spent on your thorough review of our paper. Since the author-reviewer discussion deadline is fast approaching, we kindly ask for feedback on our responses. We would be happy to discuss more if there are still some open questions.
> >
> > Best Regards,
> >
> > Authors

---

> > > ### Comment · Reviewer_1gd8 · 2024-11-25
> > >
> > > Thanks for addressing the comments and providing a very detailed set of ablations and latency experiments. The experiments clearly show the merit of the SA-Perceiver module from a latency standpoint and also show the need for both the cross and self attention between the high and low resolution image features. The remaining ablations also show the merit of the reconstructive training and the distributional alignment using pixel values and codebook indices. I have decided to increase my rating to 5 based on these.
> > >
> > > The experiment which shows the distribution of the projected visual tokens with text tokens and aligned visual tokens is a very interesting direction. However, I have the following questions regarding them:
> > >
> > > 1) How is ex1 measuring distributional alignment and how is ex2 measuring semantic alignment given that they both use the same image features taken from any image encoder like CLIP? Should it not be measuring semantic alignment in both cases as CLIP already encoded several semantic concepts.
> > > 2) The results of ex1 and ex2 seem counter-intuitive as if in ex1 the aligned visual tokens are so close in distribution to the text tokens, then shouldn't the cosine-similarity be much higher between the two in ex2? Not sure if I am missing something.
> > >
> > > I believe more analysis along these lines which shows that the training procedure taken by the authors improves the alignment of the visual tokens with the text tokens as training progresses could be a very strong indicator for the method, alongside the improved metrics on the multi-modal benchmarks. For instance, if it can be shown that compared to vanilla linear projectors/q-former architectures, if the use of the SA-Perceiver module + cognition alignment + reconstructive training leads to a faster or improved alignment between the two modalities, that would really make it a strong multi-modal training framework to be considered while training VLMs. It might be helpful to introduce some quantitative notion of this alignment and track its progress during training.

---

> > > > ### Author Response · Authors · 2024-11-26
> > > >
> > > > We sincerely appreciate your response and invaluable suggestions! Our answers to the new questions, along with additional experiments to support the enhanced alignment during the training process, are included in **1gd8_2.pdf**, which can still be found within the Supplementary Material.zip. We are looking forward to your further feedback!

---

### Official Review · Reviewer_NZ7M · 2024-11-03

**Soundness:** 2
**Presentation:** 2
**Contribution:** 2
**Rating:** 5
**Confidence:** 3

**Summary:**

The paper introduces a vision-language alignment (VLA) strategy for improving visual information extraction in multi-modal large language models. The proposed method aligns the visual input to the LLM token representations through two stages: 1) a multi-scale vision encoder trained with a reconstruction loss; and 2) an instruction-follow training with additional tasks that predict the VQ-VAE code indices and pixel RGB values. The paper integrates this strategy with LLaVA-Next and conducts experimental evaluations on 9 academic tasks and 7 MLLM benchmarks, with comparisons to previous MLLMs and an ablation study.

**Strengths:**

1. The paper addresses an important problem in the MLLMs and proposes a new two-stage vision-language alignment method.
2. As shown in the experimental evaluation, the proposed method achieves strong performances across a wide range of tasks/benchmarks, and the ablation also clearly shows the impact of each component on the final performance.

**Weaknesses:**

1. The designs of some key components in the method are not clearly motivated and lack novelty. In particular, the two-scale image representation seems to be widely used in vision encoder design. Moreover, the semantic alignment for MLLMs is not new. Several recent works have added new finetuning tasks that aim for the object or visual relation prediction to improve the cognition capability, e.g., [a] Learning by Correction: Efficient Tuning Task for Zero-Shot Generative Vision-Language Reasoning, CVPR 2024, and others.
2. The proposed method employs several pre-trained models (Stable Diffusion, VQ-VAE) and a complex architecture/training pipeline. Such a design choice lacks justification or ablation analysis. For example, it is unclear why the so-called SA-Perceiver needs to take such a structure (also see comments on ablation below). In addition, while the reconstruction loss makes sense, it is also unclear why a latent diffusion model is necessary for this loss term. Furthermore, what is the advantage of choosing VQ-VAE code indices compared to predicting objects in the finetune tasks?
3. The experimental evaluation seems lacking in the following aspects:
    - Regarding the experimental setup, the paper only considered the integration between LLaVA-Next and the proposed VLA strategy. As this method is an add-on for MLLMs, it should also report evaluations on other MLLMs to show its generality.
    - The comparisons with previous methods (except LLaVA-Next) seem unfair, as this method has access to several additional pre-trained models during training, a much better LLM (LLaMA3), and additional OCR training data. The setting alignment with LLaVA-Next seems also questionable (Line 396).
    - Table 3 shows mixed results from the ablation analysis. As the author observed, the proposed vision encoder and reconstructive training do not always work as expected.
    - The ablation study should also include comparisons with common baseline choices for the module design, including the vision encoder architecture, reconstruction process, and fine-tuning tasks.

**Questions:**

Please address the concerns raised in the weaknesses.

---

> ### Author Response · Authors · 2024-11-19
> **Response to the reviewer NZ7M**
>
> We sincerely appreciate the constructive feedback provided by the reviewer, which has been invaluable in enhancing the quality of our paper!
>
> Over the past week, we conducted the necessary experiments and thoughtfully addressed each concern raised by the reviewer. Due to system limitations that affected the proper display of our updated Figures and some Tables, we have included our responses in the Supplementary Material. Please refer to the detailed replies in **NZ7M.pdf**, which are located within the **Supplementary Material.zip**.

---

> > ### Author Response · Authors · 2024-11-23
> >
> > Dear Reviewer NZ7M,
> >
> > Thank you again for the time and effort spent on your thorough review of our paper. Since the author-reviewer discussion deadline is fast approaching, we kindly ask for feedback on our responses. We would be happy to discuss more if there are still some open questions.
> >
> > Best Regards,
> >
> > Authors

---

### Meta-Review · Area_Chair_SJmW · 2024-12-16

**Metareview:**

This paper prposes VLSA, a vision language alignment strategy for MLLM, the main contribution includes new modules in cluding perception alignment and cognition alignment. This paper recieved consistent negative socres, mainly focus on the unclear motivation  of using VAE and VQVAE  enhance the capabilities of MLLM, the overall architecture and training paradigm is quite complex and hard to generalize, hence not able to clear vilidate the effecitvieness of the method. The AC agrees with the reviewers and recommend rejection for its current version.

**Additional Comments On Reviewer Discussion:**

after rebuttal, reviewer Gehn feel that the motivation of using VAE and VQVAE can enhance the capabilities of MLLM is not clear, and all reviewers question the computational complexity, incomplete ablation studies of the methods.

---

### Decision · Program_Chairs · 2025-01-22

Reject